# Identification of Ancestry Informative Marker (AIM) Panels to Assess Hybridisation between Feral and Domestic Sheep

**DOI:** 10.3390/ani10040582

**Published:** 2020-03-30

**Authors:** Elisa Somenzi, Paolo Ajmone-Marsan, Mario Barbato

**Affiliations:** Department of Animal Science, Food and Nutrition – DIANA, and Nutrigenomics and Proteomics Research Center – PRONUTRIGEN, Università Cattolica del Sacro Cuore, Via Emilia Parmense 84, 29122 Piacenza, Italy; elisa.somenzi@unicatt.it (E.S.); paolo.ajmone@unicatt.it (P.A.-M.)

**Keywords:** hybridisation, ancestry informative marker, mouflon, sheep, random forest

## Abstract

**Simple Summary:**

Once present in the entirety of Europe, mouflon (wild sheep) became extinct due to intense hunting, but remnant populations survived and became feral on the Mediterranean islands of Corsica and Sardinia. Although now protected by regional laws, Sardinian mouflon is threatened by crossbreeding with domestic sheep causing genetic hybridisation. The spread of domestic genes can be detrimental for wild populations as it dilutes the genetic features that characterise them. This work aimed to identify diagnostic tools that could be applied to monitor the level of hybridisation between mouflon and domestic sheep. Tens of thousands of genetic markers known as single nucleotide polymorphisms (SNPs) were screened and we identified the smallest number of SNPs necessary to discriminate between pure mouflon and sheep. We produced four SNP panels of different sizes which were able to assess the hybridisation level of a mouflon and we verified that the SNP panels efficacy is independent of the domestic sheep breed involved in the hybrid. The implementation of these results into actual diagnostic tools will help the conservation of this unique and irreplaceable mouflon population, and the methodology applied can easily be transferred to other case studies of interest.

**Abstract:**

Hybridisation of wild populations with their domestic counterparts can lead to the loss of wildtype genetic integrity, outbreeding depression, and loss of adaptive features. The Mediterranean island of Sardinia hosts one of the last extant autochthonous European mouflon (*Ovis aries musimon*) populations. Although conservation policies, including reintroduction plans, have been enforced to preserve Sardinian mouflon, crossbreeding with domestic sheep has been documented. We identified panels of single nucleotide polymorphisms (SNPs) that could act as ancestry informative markers able to assess admixture in feral *x* domestic sheep hybrids. The medium-density SNP array genotyping data of Sardinian mouflon and domestic sheep (*O. aries aries*) showing pure ancestry were used as references. We applied a two-step selection algorithm to this data consisting of preselection via Principal Component Analysis followed by a supervised machine learning classification method based on random forest to develop SNP panels of various sizes. We generated ancestry informative marker (AIM) panels and tested their ability to assess admixture in mouflon *x* domestic sheep hybrids both in simulated and real populations of known ancestry proportions. All the AIM panels recorded high correlations with the ancestry proportion computed using the full medium-density SNP array. The AIM panels proposed here may be used by conservation practitioners as diagnostic tools to exclude hybrids from reintroduction plans and improve conservation strategies for mouflon populations.

## 1. Introduction

Genetic hybridisation among related species is increasingly studied due to its role in the adaptation, evolution, and diversification of species [1,2,3,4,5]. Hybridisation due to crossbreeding between interfertile species can occur due to range overlap of the populations [6], or can be promoted by anthropogenic activities [7,8,9]. Habitat degradation and livestock translocation have recently increased the rate of hybridisation events worldwide, contributing to the erosion of genetic diversity, and, in some cases, to the extinction of locally adapted wild animals [10,11]. In Europe, the genetic diversity represented by many locally adapted wild species is threatened due to hybridisation with their domestic counterpart [6]. Among these species is the European mouflon (*Ovis aries musimon*) present on the Mediterranean islands of Corsica and Sardinia [12,13]. The European mouflon is considered a remnant of the first wave of sheep domestication that occurred in the Fertile Crescent ~11,000 years ago (YA), whereas the current domestic sheep (*Ovis aries*) have been associated to a second wave of domestication which occurred ~5,000 years later [14]. The European mouflon was introduced in Corsica and Sardinia ~6-7,000 YA following human migrations and established feral populations (i.e., domesticates that have returned to the wild state) which survived until today in the harshest mountainous area of the islands [14]. Sardinia hosts the largest extant autochthonous European mouflon population [12,13,15]. Throughout the last century the Sardinian mouflon was reduced to the brink of extinction due to intense hunting and habitat erosion, until it was declared endangered and protected under local Governmental laws (Legge Regionale n23 del 1998). Since then, conservation efforts have been pursued to increase the mouflon presence in the island through translocation and genetic rescue. Since the arrival of the second wave of domestication in Europe, the sheep population brought to Sardinia has lived in sympatry with the already established mouflon population residing in the islands. Crosses between mouflon and domesticated sheep are documented since Roman times [12] and evidence of introgression from domestic sheep to mouflon has been occasionally reported by microsatellites and single nucleotide polymorphism (SNP) genotyping [12,16]. Importantly, strong signals of recent introgression were detected in an enclosed mouflon population used for restocking across the island [12]. Such an occurrence is not surprising, as the identification of hybrids based on morphological features is unreliable, especially when backcrosses occur and individuals no longer show a distinguishable intermediate phenotype between parental taxa [10]. Hence, establishing new populations from crossbred founders represents a threat for the Sardinian autochthonous genetic diversity. The use of genome-wide analysis using DNA arrays with tens or hundreds of thousands of SNPs allows the accurate detection of hybrid individuals despite confounding morphological evidence [17]. However, the large-scale use of DNA arrays can be challenging for the average financial availability of conservation projects [18].

Small sets, or panels, of ancestry informative markers (AIMs) have been developed and used to infer population genetic parameters in several species. AIM panels have been used to estimate biogeographical ancestry and structure in human populations [19,20,21,22], to discriminate among breeds and geographical origin of Italian sheep breeds [23], to identify cattle breeds [24,25,26], to trace the origin of animal products [27,28], and for breed assignment and analysis of individual ancestry in cattle [29,30,31,32] and pig populations [33]. Moreover, AIM panels have been developed to identify hybrids in wildlife conservation projects on wolf [34], wild cat [9], and mule deer [35].

In this work, we applied supervised machine learning approaches on mid-density DNA array data and identified AIMs able to correctly discriminate mouflon *x* domestic sheep crosses when tested both on real and simulated data. Our results provide a fundamental conservation tool for the affordable identification of hybrid mouflon *x* domestic sheep and for the quantification of their admixture level.

## 2. Materials and Methods 

We collected publicly available genotype data (~50k SNPs) from 23 non-admixed feral Sardinian mouflon (MSar) and 23 Sarda sheep (SAR) (Appendix A), and from 28 Sardinian mouflon hybrids (MxS) showing extensive levels of admixture according to previous analyses [12,36]. The Sarda sheep is an autochthonous Sardinian sheep breed counting almost four million heads in Sardinia. It is reared for its high milk production and accounts for almost all of the sheep presence in the island. Additionally, we collected genotypic data from 26 Lacaune (LAC), 28 Australian Poll Merino (MER), and 23 New Zealand Texel (TEX) made available by the Sheep HapMap project [37]. Lacaune is a milk-producing breed which has been recently imported into Sardinia, whereas Merino and Texel are two cosmopolitan breeds reared for wool and meat production, respectively [37].

Non-autosomal variants, markers with missing data (> 0), minor allele frequency (MAF) ≤ 0.1, and markers significantly out of Hardy–Weinberg equilibrium (HWE ≤0.001) were excluded from the subsequent analyses. Pruning was performed using PLINK v1.9 [38].

### 2.1. Aims Identification and Panel Development

To identify the markers from the medium-density SNP array genotyping data for MSar and SAR able to detect admixture between mouflon and domestic sheep, we applied a two-fold approach which included (1) a preselection step aimed at removing the least significant markers, and (2) supervised machine learning classification approaches to identify the most informative markers among those left after preselection.

### 2.2. Preselection 

The dataset was first reduced based on Principal Component Analysis (PCA) results. PCA is a widely used dimensionality reduction technique that generates new uncorrelated variables (principal components; PCs) sorted according to the variance they explain. The contribution of each SNP to every PC is expressed as a loading score. A PCA analysis of the reference breeds was performed using R v3.4.3 [39]. The first PC discriminated mouflon against domestic sheep, whereas the second PC described within-mouflon structure (see Results). Similarly, the following PCs identified within-population substructure. Consequently, we only considered the first PC. The loading scores of the first PC were squared (see [21,24]), and those SNPs associated with loading values exceeding 1.5x the interquartile range of the loading distribution were retained for further analyses (hereon: GW1; Appendix A).

### 2.3. Supervised Classification

SNPs which passed preselection were submitted to supervised machine learning classification, as implemented in Boruta v6.0.0 [40], in order to select the most discriminant markers. This algorithm is based on a random forest learning process that allows, after a training step, to derive a set of rules for data classification. The Boruta algorithm measures the importance of an attribute (here, a SNP) through the loss of classification accuracy due to random permutation of the attribute values between objects [40]. Eventually, Boruta assigns the binary classification “confirmed important” and “confirmed unimportant” to each of the features tested. A single run of the Boruta algorithm was performed genome-wide to identify a first subset of significant SNPs (GW1) and those SNPs listed as “confirmed important” were assembled as a GW2 panel. To obtain an even smaller AIM panel, we performed 20 independent iterations of the classification algorithm on the GW1 panel; those SNPs listed as “confirmed important” in at least 19 scans out of 20 were pooled in a third panel (GW3) (Appendix A).

With the aim to identify AIM panels which forcibly intercepted all autosomes, we applied the iteration algorithm to each autosome separately (Chromosome Wide, CW). Feature selection was performed for each autosome and SNPs listed as important were assembled in a panel named CH1. Lastly, to further reduce the number of significant SNPs, a last panel (CH2) was generated by selecting from CH1 the three most discriminant SNPs per each autosome.

### 2.4. AIM Panel Validation 

The accuracy of each of the five AIM panels (three genome-wide and two chromosome-wide) to assess ancestry levels, or levels of admixture, was tested using genotype data from both simulated and real hybrid populations (MxS). To test the AIM panels with simulated hybrids, we developed Hybridiser v0.1, an R script able to simulate F1 hybrid individuals (available at https://github.com/barbatom/Hybridiser). Given two parental populations, Hybridiser computes the allele frequency at each locus, and then generates hybrid genotypes at each locus by selecting an allele from each of the parental populations with probability equal to the parental allele frequency. A dataset of 270 simulated hybrids (HYBs) was generated pooling 90 MSar *x* SAR first-generation crosses (F1) and the reciprocal backcrosses F1 *x* MSar (BC1M), and F1 *x* SAR (BC1S), each comprising 90 individuals. 

Admixture v1.3.0 [41] was used to perform supervised clustering tests to evaluate the ancestry proportions of both simulated and real hybrids using the pure mouflon and domestic populations as ancestry sources. To measure how well the AIM panels estimated the admixture level compared to that determined by the full set of markers, we compared the admixture results from each of the AIM panels (selected markers) with those from the full medium-density genotype array (full set of markers) using the coefficient of determination (*r^2^*).

To test if the AIM panels performed better than an equally sized set of SNPs chosen at random, we generated 5,000 random AIM sets, and for each random set we performed supervised admixture analysis. Finally, we computed coefficients of determination values between the ancestry assignment of the full set and the reduced random panel. Being computationally challenging, this test was performed on the smallest AIM panel set we generated (GW3). The coefficient of determination values obtained using the 5,000 random SNP sets were standardised by z-scores. An empirical *p-*value for the correlation obtained using the GW3 panel was calculated according to Davison and Hinkley [42] as *p = (1 + x)/(1 + n)*, where *x* is the number of random sets that produced an *r^2^* score greater than or equal to that calculated using GW3 and *n* is the total number of random set tested (Appendix A). Significance was set for *p*-values lower than 0.01.

Finally, we evaluated the ability of the five AIM panels to detect mouflon hybrids with domestic sheep breeds other than Sarda. For each mouflon *x* domestic sheep population, we generated a dataset of simulated hybrids composed of: i) 90 F1 offspring between MSar and the test domestic breed, ii) 90 backcrosses between F1 and the test domestic breed and iii) 90 individuals obtained as backcross between F1 and MSar. The AIM panels were tested on simulated hybrid populations of MSar with MER, LAC, and TEX, respectively. Coefficients of determination were computed between the supervised admixture results obtained using AIM panels (selected markers) and the full medium-density genotype array (full set of markers). 

## 3. Results

Pruning for missing data, MAF, and HWE left 33,481 non-rare, neutral SNPs. PCA was performed on the two reference populations (MSar and SAR) to evaluate how many PCs contributed in discriminating mouflon and domestic sheep. As expected, PC1 (18.7% of the total variance), split mouflon and domestic sheep as two distinct clusters (Figure 1, Appendix A). PC2 (7.07% of the total variance) identified subpopulation structure in mouflon exclusively (Figure 1). Consequently, PC1 was considered the only relevant component for the preselection step.

### 3.1. AIM Identification and Panel Development

The PCA-based preselection step identified 1,279 SNPs that contributed the most to discriminate between mouflon and domestic sheep reference populations (GW1; Table 1, Appendix A). This first panel was submitted to both a single run and 20 iterations of random forest selection that identified 131 (GW2), and 51 SNPs (GW3), respectively (Table 1). A single run of random forest was applied chromosome-wide and identified 933 SNPs (CH1), with a range of 10 to 73 SNPs per chromosome (Appendix A). The selection of the three most significant SNPs per chromosome from CH1 selected 78 SNPs (CH2) (Table 1).

### 3.2. Panels Validation

PCA analyses were used to visually compare the performance of the full set of SNPs and the five AIM panels applied to simulated (HYB) and real hybrid (MxS) populations. 

The PCA of HYB using the full set of SNPs discriminated the parental populations (SAR and MSar) at opposite sides of the graph and positioned the hybrid populations according to their ancestry proportions, with F1 at the centre of the plot and the two backcrosses BC1S and BC1M closer to SAR and MSar, respectively (Figure 2; Appendix A). PCA of MxS using the full set of markers identified four individuals overlapping with the pure ancestry mouflon cluster, while the others were distributed along a gradient between MSar and SAR (Figure 3; Appendix A).

PCA performed on HYB using the AIMs showed clustering comparable with the PCA results obtained using the full set of SNPs (Figure 2; Appendix A). Each AIM panel was able to discriminate the simulated ancestry proportions and showed distinct clusters for each simulated hybrid population, with only minimal cluster overlap for three individuals when using the smallest SNP set (GW3; Figure 2). When applied to the real population (MxS), GW1, GW2, GW3 and CH1 discriminated the pure and admixed individuals (Figure 3; Appendix A), whereas CH2 showed some overlap occurring between the pure mouflon and the admixed individuals having the smallest amount of domestic ancestry (Appendix A).

A quantitative assessment of the ancestry proportions in HYB was obtained through supervised Admixture using MSar and SAR as reference populations. HYB showed ancestry proportions coherent with the expected values of each offspring group (Appendix A); mean and standard deviation were 0.497 ± 0.0059 for F1, 0.234 ± 0.0059 for BC1S, and 0.76 ± 0.005 for BC1M. Supervised Admixture on MxS using the full set highlighted the heterogeneity of ancestry proportions in this population, and the presence of four individuals showing pure ancestry (Figure 4).

Coefficients of determination were then calculated between the ancestry proportions obtained using the full set of SNPs and AIM panels. The coefficients of determination were high overall across all panels (*r^2^* ≥ 0.960), with GW1 and CH2 scoring the highest and lowest, respectively (Table 2). As expected, values were proportional to the number of AIMs in the panels. In simulated individuals, GW1 scored *r^2^ =* 0.990 and CH1 *r^2^ =* 0.989, while MxS had *r^2^ =* 0.997 for both the panels. Coefficients of determination were slightly higher for AIMs identified using a genome-wide approach instead of chromosome-wide. The genome-wide GW3 panel performed better than the chromosome-wide CH2 panel despite the lower number of SNPs. AIM panels tested on HYB recorded higher correlation values compared to MxS. 

We further tested the ancestry assignments obtained using GW3 (the smallest panel among the AIMs generated) against the null hypothesis that the same results could be obtained using any equal sized set of SNPs chosen at random. The null hypothesis was rejected with high significance (*p-*value <0.001; Appendix A). 

AIM panels were further tested to detect admixture between mouflon (MSar) and three commercial sheep breeds (TEX, APM, and LAC). Coefficients of determination calculated between the ancestry percentage obtained using the AIM panels and full set of SNP ancestry resulted in *r^2^* > 0.92 (Table 3).

## 4. Discussion

Hybridisation between wild species and their domestic counterpart may lead to genetic homogenization and loss of local adaptation [10]. Sardinian mouflon is the largest extant autochthonous European mouflon population and represents a unique genetic heritage threatened by hybridisation with domestic sheep [12,15,16].

In this work we implemented a supervised machine-learning-based classification approach to identify highly discriminant markers which can be included in rapid and low-cost diagnostic assays with the aim to support mouflon conservation [43]. The accuracy of AIM panels to detect introgression depends on the quality and sample size of the reference populations, with increasing probability of capturing most of the existing within-population variability when purer/non-admixed and larger reference datasets are used [44]. To the best of our knowledge, previous work on AIM panel discovery performed on humans, domestic and wild animals always accounted >100 reference samples [19,22,23,24,26,27,34,35]. Due to the large genetic distance occurring between sheep and mouflon, we achieved reliable feature selection using a sample size of 23 individuals for each reference population. Our results showed that the AIMs we identified can accurately discriminate Sardinian mouflon ancestry from other cosmopolitan sheep breeds as well. As expected, panels counting a higher number of SNPs, such as GW1 and CH1, resulted in better performances in hybrid identification than other panels. However, previous research showed that reliable results in individual assignment can be achieved also with a number of SNPs lower than 100 [21,27,29,32,43,45,46]. Accordingly, we obtained positive results also with less than 100 AIMs, as shown by tests performed with GW3 and CH2 panels. We tested AIM identification using both chromosome-wide and genome-wide approaches as done in previous research (see [45]). Our results confirmed the genome-wide approach to perform better than chromosome-wide. Chromosome-wide selection may introduce feature redundancy by selecting several SNPs on different chromosomes but carry identical discriminant power. Conversely, the genome-wide approach might be better at selecting highly discriminant SNPs which also capture a wider range of the focal population diversity. Noticeably, the distribution of AIMs in the genome-wide approach appears to be not homogeneous along the genome. Indeed, two phylogenetically close references, as in this case, are likely to share most of the genome, with few genomic regions responsible for much of the genomic distance, further supporting the genome- over the chromosome-wide approach as best suited to intercept such regions.

We assessed the performance of the panels in identifying crosses between domestic sheep and feral mouflon using both simulated and real data. As expected, using the AIMs on simulated data performed moderately better than on real admixed mouflon samples. Real admixed populations present a more complex genetic make-up, influenced by demography, selection, inbreeding and introgression events occurred during centuries. Conversely, the simulated individuals were generated from the same reference populations used to select the best AIMs. In addition, the mating system applied in simulations generates simplified admixture patterns with respect to those occurring in real populations. However, the use of simulated hybrid genotypes may aid in AIM panels testing to overcome the lack of real admixed samples. When available, real hybrids samples can guarantee the accuracy of the AIMs panels.

## 5. Conclusions

The detection of hybrids is a fundamental task in biodiversity conservation, and genetics provides an unmatched tool to identify even cryptic levels of hybridization. Applying a machine learning-based approach, we identified reduced SNP panels which proved effective in identifying domestic sheep introgression in Sardinian mouflon. The AIM panels we propose can be applied for large-scale assessment of the Sardinian mouflon population, and aid in the conservation of this unique genetic resource. Lastly, the method we present can be easily applied to other wildlife and domestic species conservation to develop accurate and affordable tools for hybrid identification.

## Figures and Tables

**Figure 1 animals-10-00582-f001:**
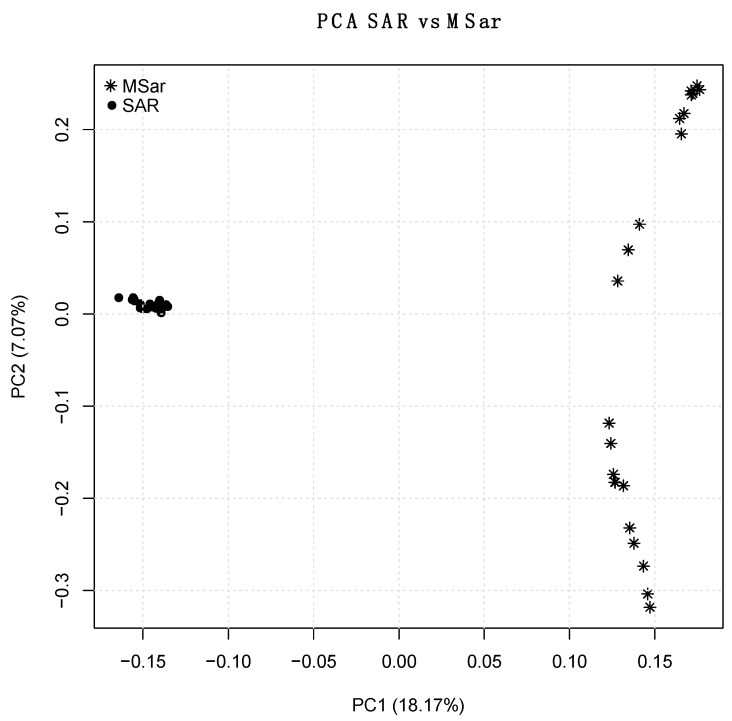
Principal Components Analysis (PC1 vs. PC2) of the two reference populations (MSar and SAR) analysed using the full single nucleotide polymorphism (SNP) set. In brackets are the percentage of variance explained by each component.

**Figure 2 animals-10-00582-f002:**
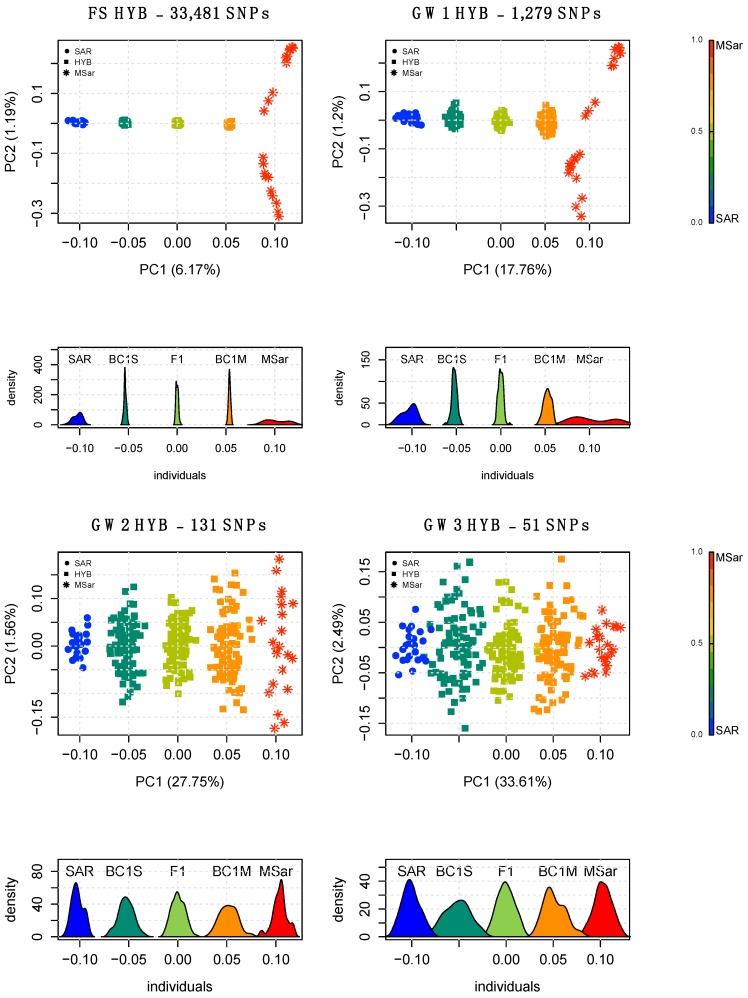
PCA and density distribution of the PC1 obtained using the full SNP set (top-left panel) and three AIMs on reference populations and simulated hybrids (HYB) using the genome-wide discovery approach. BC1S and BC1M are the simulated F1 backcrossed with SAR and MSar, respectively. The gradient legend on the right side of the plot shows the transition gradient from sheep to mouflon genetic components.

**Figure 3 animals-10-00582-f003:**
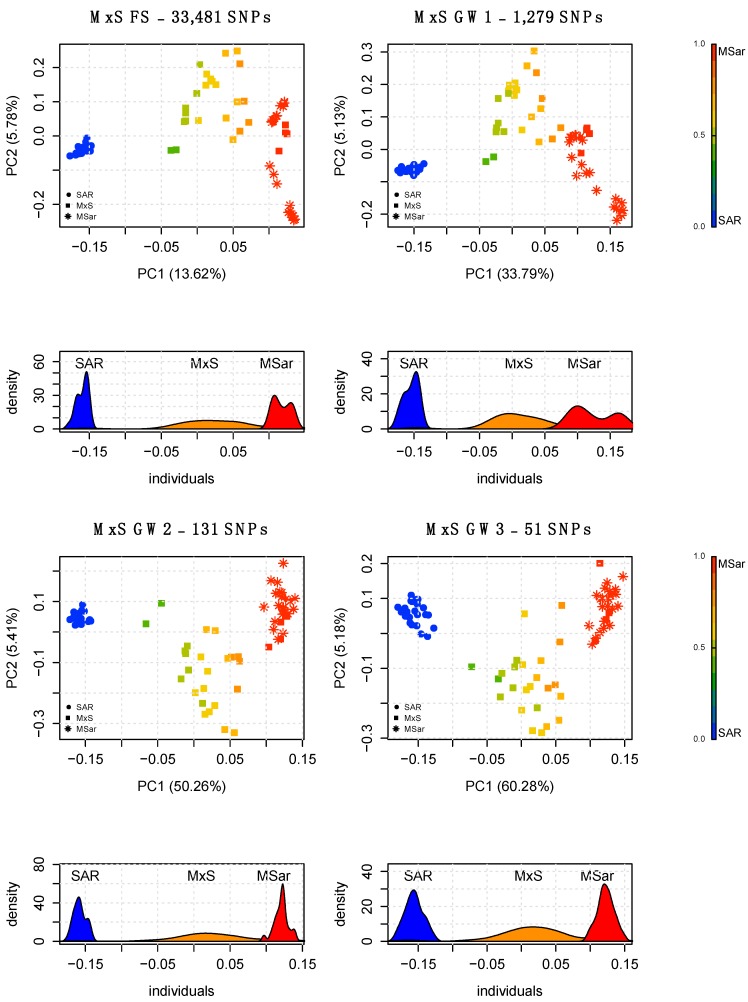
PCA and density distribution of the PC1 obtained using the full SNP set (top-left panel) and three AIMs on reference populations and real mouflon x domestic hybrids (MxS) using the genome-wide discovery approach. The gradient legend on the right side of the plot shows the transition gradient from sheep to mouflon genetic components. The analysis was performed using 33,481 SNPs and the three GW panels.

**Figure 4 animals-10-00582-f004:**
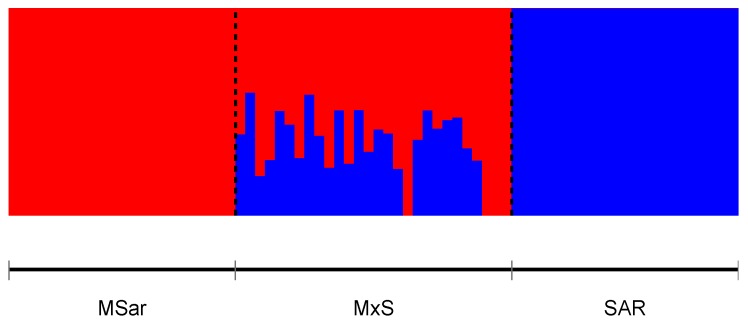
Supervised Admixture plot of MxS dataset obtained using the full set of SNPs. MSar and SAR were used as prior populations.

**Table 1 animals-10-00582-t001:** Characteristics of the ancestry informative marker panels. The data processes used for marker selection were: preselection (Pre), Random Forest (RF), iterated Random Forest (iRF), and top-markers choice (tc). N is the number of SNPs in each panel. The SNP distribution per chromosome can be found in Appendix A.

Panel Name	Scope	Method	N
GW1	Genome-wide	Pre	1279
GW2	Genome-wide	Pre + RF	131
GW3	Genome-wide	Pre + RF + iRF	51
CH1	Chromosome-wide	Pre + RF	933
CH2	Chromosome-wide	Pre + RF + tc	78

**Table 2 animals-10-00582-t002:** Coefficient of determination values (r^2^) calculated between the ancestry percentages using the full set of SNPs and the AIM panels in the simulated (HYB) and case study (MxS) populations. N is the number of SNPs in each panel.

AIMs	N	HYB	MxS
GW1	1279	0.997	0.99
GW2	131	0.985	0.971
GW3	51	0.966	0.966
CH1	933	0.997	0.989
CH2	78	0.961	0.946

**Table 3 animals-10-00582-t003:** Coefficient of determination values calculated between the ancestry percentages obtained using the full SNP set and the AIMs in three commercial sheep breeds.

Breed	Acronym	GW1	GW2	GW3	CH1	CH2
New Zealand Texel	TEX	0.995	0.971	0.945	0.993	0.920
Australian Poll Merino	APM	0.995	0.968	0.938	0.993	0.923
Lacaune	LAC	0.994	0.962	0.928	0.992	0.923

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
