# Peer review of "Identification of Ancestry Informative Marker (AIM) Panels to Assess Hybridisation between Feral and Domestic Sheep"

_animals, 2020, doi:10.3390/ani10040582_

Round 1

Reviewer 1 Report

This is a significant piece of research that will contribute to the conservation of an endangered species.

Comment and suggestion are on the attached pdf.

Author Response

We are extremely grateful to reviewer 1 for the detailed revision and for pointing out many sentences which needed to be clarified.

Title: replace "feral" with "wild" to be consistent with the rest of the manuscript.

Answer: We appreciate the reviewer point in being consistent with the wording. Hence, we decided to keep the term “feral” rather than “wild” throughout the manuscript as it is more accurate when referring to Sardinian mouflon (and consequently: European mouflon), as this population was first domesticated and then went back to feral life . We kept the term “wild” in the manuscript exclusively when explaining hybridisation in general terms.

Line 150: This sentence needs to be clearer. For example:

The admixture results from each of the AIM panels (selected markers) and from the full medium density genotype array (full set of markers) were compared using linear regression. The coefficient of determination (r-squared, r2) was used to measure of how well the AIM panels estimated the admixture level compared to that determined by the full set of markers.

Answer: The suggested sentence has been implemented in the text with a more detailed description of the process.

Line 157: More explanation needed. What level of significance (Alpha value) was set?  At a significance level of alpha = 0.05, in 5000 correlations means that you would expect 250 significant test by chance....

How were the coefficients of determination compared? Was a one-tailed test used as the hypothesis would be that the random marker set would have lower coefficients of determination values...

Answer: The relevant section has been amended providing an explanation of the formula used to generate the p-value and the significance level.

Line 164: This need to be written more clearly. See comment above about the use of the coefficient of determination.

Answer: The sentence has been improved explaining how the coefficient of determination was used for comparing admixture results.

Table 1: By convention table usually only have three horizontal lines: One at the top and bottom and one underneath the column labels.

Answer: All the table were formatted accordingly.

Line 203: The CH1 and 2 data would be better as part for Figures 2 and 3 as opposed to supplementary figures

Answer: We see the reviewer point. However, after careful consideration (and many attempts) we opted to not merge the CH figures with Fig 2 and 3 as it seems to us that such solution worsens the readability of the figures.

Line 250: This paragraph needs to be written more clearly. Are the authors trying to say that the GW3 coefficient of determination values for each of the population types were significantly higher than those of the randomly selected marker sets?

Answer: The paragraph has been improved and clarified.

Line 275: This is a discussion of the results and therefore does not require in-text citations referring back to the results section

Answer: All in-text citations to Results were removed from Discussion.

Line 307: Reference citations are not appropriate in a conclusion

Answer: Reference citations were removed from Conclusions.

All the other suggestions and corrections (typos, etc.) were accepted and implemented in the manuscript as suggested, as highlighted by Track Change.

Reviewer 2 Report

Dear Authors,

it was a pleasure to read your Manuscript I hope to see it "in press" as soon as possible.

I have only one suggestion - in M&M section I have not found the explanation of what data is PC2 discriminating. 

Otherwise, the study is comprehensive and scientific important.

Author Response

We thank reviewer 2 for seeing merit and supporting our work

Comment: In M&M section I have not found the explanation of what data is PC2 discriminating.

Answer: A brief description of PC2 has been included in M&M.